# Investigating Connectivity Gradients in Schizophrenia: Integrating Functional, Structural, and Genetic Perspectives

**DOI:** 10.3390/brainsci15020179

**Published:** 2025-02-11

**Authors:** Jie Xiang, Chengze Ma, Xiuhui Chen, Chen Cheng

**Affiliations:** 1College of Computer Science and Technology (College of Data Science), Taiyuan University of Technology, No. 79 Yingze West Street, Taiyuan 030024, China; 2Department of Neurology, Max Planck Institute for Human Cognitive and Brain Sciences, 04103 Leipzig, Germany

**Keywords:** functional connectivity gradients, morphological similarity gradient, gene expression

## Abstract

**Background:** Schizophrenia is a complex disorder characterized by disruptions in cognition, behavior, and emotions. Extensive research has uncovered alterations in a single modality (either the brain structure or function) in schizophrenia. However, the limitation is that a single modality could not offer a synchronous result between the brain structure and function because of different samples. Here, a multiparametric approach is essential to understand the common and distinct alterations between the brain structure and function in schizophrenia. **Methods:** We analyzed structural and functional magnetic resonance imaging data from 146 participants (72 individuals with schizophrenia and 74 healthy controls). Individual morphological similarity and functional connectivity gradients were computed using a nonlinear dimensionality reduction technique with diffusion map embedding. Furthermore, to understand how the alterations may be related to genetic underpinnings, gene expression enrichment analyses were conducted using Allen Brain Human Atlas and GOrilla. **Results:** Compared with controls, patients with schizophrenia had reduced scores on the principal functional gradient of the visual network and elevated scores on the principal functional gradient of the limbic network, the frontoparietal control network, and the default mode network. Additionally, the main functional gradient in individuals with schizophrenia showed compression along the primary axis compared to the healthy control group. These changes were linked to genes involved in synaptic signaling and neuronal development. **Conclusions:** These results indicate connectome gradient dysfunction in schizophrenia and its linkage with gene expression profiles, supporting widespread network-level abnormalities. The integration of neuroimaging provides insight into the neurobiological underpinnings and potential biomarkers for treatment evaluation in this disorder.

## 1. Introduction

Schizophrenia is a chronic and severe mental disorder characterized by cognitive, behavioral, and emotional dysfunctions [1]. It is marked by symptoms such as hallucinations, delusions, disorganized thinking, and impaired cognitive abilities, which significantly impact the quality of life of affected individuals [2,3,4]. For decades, research has endeavored to elucidate the neural biomarkers underlying schizophrenia, yet these biomarkers remain incompletely understood. One critical reason is that the link between the brain structure and function remains unclear. Although converging evidence in schizophrenia points to widespread alterations in the brain structure and function [5,6,7], how these alterations are linked to each other remains to be studied.

Regarding the functional brain, connectivity gradients have emerged as a powerful framework for understanding the hierarchical organization of brain networks, offering novel insights into the global architecture of functional connectivity [8,9,10]. These gradients provide a low-dimensional embedding of functional connectivity patterns, capturing transitions from unimodal sensory regions to transmodal association areas. Alterations in these gradients have been implicated in several psychiatric disorders, including schizophrenia, where they may reflect disruptions in the integration and segregation of brain networks [11,12]. Studies have shown that in individuals with schizophrenia, there is an altered hierarchy of brain functional gradients, particularly affecting the primary-to-transmodal axis, which results in a disrupted balance between sensory and associative regions [13,14]. Additionally, reduced gradient variance has been found, indicating less distinct transitions between different functional systems, thereby highlighting compromised functional network integration in schizophrenia.

Similarly, morphological similarity networks (MSNs) offer insights into the structural organization of the brain by capturing correlations between morphological features across regions [15,16,17]. These networks have been used to investigate structural connectivity alterations in schizophrenia, highlighting abnormal patterns in the cortical thickness, surface area, and other morphological characteristics [18,19]. Studies have shown that individuals with schizophrenia exhibit reduced MSN connectivity, particularly between association cortices, which suggests the disrupted integration of higher order cognitive regions. For example, Nenadic et al. [20] reported a significant decrease in morphological similarity in the frontal and temporal cortices, which correlated with the cognitive impairments observed in schizophrenia. Additionally, altered MSN topology has been linked to more severe symptoms and may serve as a structural marker of disease progression [21]. By combining functional and structural gradient analyses, it is possible to gain a more comprehensive understanding of the neural alterations underlying schizophrenia [22,23].

Nevertheless, genetic factors play a critical role in shaping the brain structure and function, as well as the pathophysiology of schizophrenia. Genome-wide association studies (GWASs) have identified numerous genetic loci associated with the disorder, many of which are related to synaptic function, neuronal development, and neurotransmission [24,25,26]. Recent studies have demonstrated that genetic variants linked to schizophrenia are correlated with cortical thickness and gray matter volume reductions, particularly in brain areas involved in cognitive processing and executive function [27,28]. For example, Gandal et al. [29] found that dysregulated gene expression in schizophrenia is largely associated with synaptic and mitochondrial pathways, which in turn may affect the integrity of neural networks crucial for cognitive function. Additionally, Paquola et al. [16] reported that the spatial patterns of gene expression are aligned with the regions of abnormal morphological similarity networks in schizophrenia, suggesting a genetic basis for structural brain abnormalities. Integrating gene expression data with neuroimaging measures enables researchers to link genetic risk factors with observed alterations in brain connectivity, providing valuable insights into the molecular mechanisms underlying schizophrenia [30].

The current study aimed to investigate the differences in functional and morphological connectivity gradients between individuals with schizophrenia and healthy controls. We adopted a multiparametric approach that integrated functional MRI (fMRI), structural MRI (sMRI), and gene expression data to explore altered brain connectivity in schizophrenia. By examining the relationship between gene expression profiles and connectivity gradients, this study sought to elucidate the molecular mechanisms contributing to altered brain organization in schizophrenia, thereby advancing our understanding of the disorder’s neurobiological basis. Importantly, understanding these connectivity alterations may have significant clinical implications; for example, identifying specific molecular and neuroimaging biomarkers associated with altered brain connectivity could potentially improve the accuracy of schizophrenia diagnosis, allowing for earlier intervention. By linking molecular mechanisms to large-scale brain connectivity patterns, this research may aid in identifying therapeutic targets and optimizing intervention approaches for schizophrenia.

## 2. Materials and Methods

### 2.1. Participants

The Center for Biomedical Research Excellence (COBRE; https://coins.trendscenter.org/, accessed on 25 June 2024) contributed raw anatomical and functional MR data from 72 patients with schizophrenia and 74 healthy controls (ages ranging from 18 to 65 in each group). All subjects were screened and excluded if they had a history of neurological disorders, history of intellectual disability, history of severe head trauma with more than a 5 min loss of consciousness, or history of substance abuse or dependence within the last 12 months. Refer to Table 1 for the dataset specifications.

### 2.2. Data Acquisition

The MRI parameters were set at TE = 29 ms and TR = 2000 ms, with a slice size of 64 × 64 and 32 slices per volume. The field of view (FOV) was 192 mm, and the voxel size was 33 × 4 mm^3^. Each subject underwent a 5 min resting-state run (150 TR). Furthermore, the general research strategy of this article is shown in Figure 1.

### 2.3. Image Preprocessing

#### 2.3.1. Functional Data Preprocessing

For functional data, the initial 10 volumes were discarded to allow for scanner stabilization and subject acclimatization. The remaining volumes were preprocessed using the DPABI_V8.2 toolbox [31]. Preprocessing included slice timing correction, head motion correction, spatial normalization to the Montreal Neurological Institute (MNI) standard template (resampled to 3 × 3 × 3 mm^3^), spatial smoothing with an 8 mm full width at half maximum (FWHM) Gaussian kernel, linear drift removal, and band-pass filtering (0.01–0.08 Hz) to reduce low-frequency drift and high-frequency noise. Following preprocessing, average time series were extracted from each region of interest (ROI), and ROI time point signals were normalized.

#### 2.3.2. Structural Data Preprocessing

For T1-weighted imaging, T1-weighted images were preprocessed using the FreeSurfer V7.4.1 (https://surfer.nmr.mgh.harvard.edu, accessed on 25 June 2024) recon-all pipeline. The pipeline includes skull stripping, tissue segmentation, surface reconstruction, metric reconstruction, and spherical normalization.

#### 2.3.3. Gene Expression Dataset and Preprocessing

The Allen Human Brain Dataset (AHBA) provides comprehensive brain coverage, with gene expression data from six postmortem brains comprising 3702 samples. We utilized the abagen V0.1.3 toolbox (https://www.github.com/netneurolab/abagen, accessed on 25 June 2024) [32,33,34] to process the transcriptome data and map them to 400 brain regions. Preprocessing included probe-to-gene annotation updates, intensity-based filtering, probe selection, sample-to-region matching, the handling of missing data, sample normalization, gene normalization, the computation of sample-to-region metrics, and the selection of stable genes.

### 2.4. Analysis of Functional Connectivity Gradients

For each subject, we constructed a 400 × 400 ROI-based functional network using the Schaefer 400 parcellation atlas [35]. The BrainSpace toolbox (https://brainspace.readthedocs.io/en/latest, accessed on 25 June 2024) [36] was used to generate ROI-based functional connectivity gradients from preprocessed fMRI data. The top 10% of connections for each ROI represent the most typical connections between that ROI and other ROIs, and weak connections due to noise can be removed. Therefore, the cosine similarity between ROI pairs was computed using the top 10% of connections of each ROI [37]. Diffusion map embedding was applied to capture gradient components explaining differences in functional connectivity patterns. The gradient maps were aligned across individuals using iterative Procrustes rotation. Given the importance of gradients related to the neuronal microstructure and cognitive function, our focus was on alterations in the principal gradients associated with schizophrenia. To visualize the functional axis captured by each gradient, we performed Neurosynth [38] decoding on the group gradient maps. Further, we calculated the average gradient score for all parcels within each of the seven connectivity networks described by Yeo and colleagues [39]. Significant results were set at a Benjamani–Hochberg false discovery rate (FDR)-corrected q < 0.05.

### 2.5. Analysis of Morphological Similarity Network (MSN) Gradients

For T1-weighted images processed using FreeSurfer, Schaefer 2018 clusters were generated in the volume space, projected onto individual cortical surfaces, and used to calculate anatomical metrics. Five structural features—the gray matter volume (GM), cortical thickness (CT), surface area (SA), intrinsic curvature (IC), and mean curvature (MC)—were extracted. The morphological features for each subject were z-score-normalized over 400 regions, and Pearson correlation coefficients were calculated between pairs of z-score-normalized morphological vectors to form a 400 × 400 MSN matrix for each subject. Morphological similarity gradients were constructed using the BrainSpace toolbox.

### 2.6. Association Analysis of Gene Expression and Altered Gradients

Partial least squares (PLS) regression analysis was used to investigate relationships between transcriptional profiles and intergroup differences in the main gradient maps. We performed a *t*-test on the gradient values of both in 400 brain regions, calculated the t-values, and then normalized the t-values using the z-score to obtain the z-map. Analysis of gene expression data and z-map of gradient differences between groups. The gene expression data and z-maps served as the predictor and response variables, respectively. We obtained the first two principal components of the partial least squares. A spatial autocorrelation-corrected permutation test was employed to determine if the R² of the PLS component exceeded chance levels. Following the identification of significant components, bootstrap methods were used to correct the estimation errors for gene weights. Genes were ranked based on the corrected weights, indicating their contributions to the PLS components. Enriched gene ontology (GO) terms were identified using GOrilla (https://cbl-gorilla.cs.technion.ac.il/, accessed on 25 June 2024) across three categories: biological processes, molecular functions, and cellular components. Significant enrichment was determined at a q < 0.05 after Benjamani–Hochberg false discovery rate (FDR) correction.

## 3. Results

The principal functional gradient explained 30.5 ± 5.2% of the total connectivity variance (schizophrenia: 30.3 ± 5.3%; controls: 30.7 ± 5.2%). The secondary gradient from the visual network to the sensorimotor network explained 17.8 ± 5.0% of the total connectivity variance (schizophrenia: 17.6 ± 4.8%; controls: 18.0 ± 5.2%). For the principal morphological similarity gradient (MSN), 25.2 ± 2.3% of the total variance was explained (schizophrenia: 25.6 ± 3.0%; controls: 24.6 ± 1.1%). The secondary gradient of the morphological similarity gradient (MSN) explained 18.4 ± 2.5% of the total connectivity variance (schizophrenia: 18.2 ± 3.1%; controls: 18.6 ± 1.4%) (see Appendix A, for the above results). The first two components retained a large number of differences while retaining a small number of components; therefore, we chose the first two gradients to conduct the subsequent analysis.

### 3.1. Macroscale Cortical Gradients

The first two group-level gradients are shown in Figure 2, and their Neurosynth meta-analytic associations and relationships to the Yeo networks [36] are shown in Figure 3 (seven-network solution). The comparison of functional gradients and morphological similarity gradients between the normal control and schizophrenia groups revealed distinct patterns. The main functional gradient in individuals with schizophrenia showed compression along the primary axis compared to the normal control group, while no significant differences were observed in the morphological similarity gradients between the two groups (Figure 2a). Similarly, the secondary gradients of the functional and morphological similarity gradients showed a consistent pattern, with the schizophrenia group exhibiting compression but no notable differences in the morphological similarity gradients (Figure 2b).

When analyzing the seven Yeo functional networks (https://doi.org/10.1152/jn.00338.2011, accessed on 25 June 2024), significant differences in the functional gradients were found between individuals with schizophrenia and controls. Regarding the primary gradient of the functional gradient, the limbic network (t = −9.786, *p* < 0.05), the frontoparietal control network (t = −5.538, *p* < 0.05), and the default mode network (t = −8.773, *p* < 0.05) exhibited higher functional gradient scores in the schizophrenia group compared to the control group, whereas the visual network (t = 7.582, *p* < 0.05) showed lower functional gradient scores in individuals with schizophrenia relative to healthy controls (Figure 3a). In terms of the secondary gradient of the functional gradient, the sensorimotor network (t = 9.933, *p* < 0.05), the limbic network (t = 4.157, *p* < 0.05), the frontoparietal control network (t = 7.747, *p* < 0.05), and the default mode network (t = 11.118, *p* < 0.05) showed lower scores in the functional gradient in individuals with schizophrenia compared to controls (Figure 3b). Significant differences were also found between individuals with schizophrenia and controls on the morphological similarity gradient. In terms of the principal gradient of the functional gradient, the sensorimotor network (t = 2.697, *p* < 0.05) in individuals with schizophrenia showed lower gradient scores compared to controls. In terms of the secondary gradient of the morphological similarity gradient, the visual network (t = −4.554, *p* < 0.05) in individuals with schizophrenia showed higher gradient scores compared to controls (see Appendix A).

To justify the sample size, we performed a power analysis on the gradient results, and the results of the analysis were as follows: For the principal gradient of the functional gradient, the effect sizes of the networks were as follows: limbic network: −1.635; frontoparietal control network: −0.925; default mode network: −1.465; and visual network: 1.266. The statistical validity of each network was as follows: limbic network: 1.000; frontal control network: 1.000; default mode network: 1.000; and visual network: 1.000. For the secondary gradient of the functional gradient, the effect sizes for each network were as follows: sensorimotor network: 1.66; limbic network: 0.695; frontoparietal control network: 1.294; default mode network: 1.856. The statistical validity of each network was as follows: sensorimotor network: 1.000; limbic network: 0.986; frontoparietal control network: 1.000; default mode network: 1.000. For the principal gradient of the morphological similarity gradient, the effect size for the sensorimotor network was 1.629 and the statistical validity was 1.000. For the secondary gradient of the morphological similarity gradient, the effect size of the visual network was 0.926 and the statistical validity was 1.000 (see Appendix A). Overall, the functional and morphological gradients had strong statistical validity across all major networks. The results of these analyses suggest that the sample size was appropriate because the strong statistical validity and significant effect sizes across all major networks, particularly for the main gradients of the functional data, adequately indicate that these networks responded significantly to the gradients examined. Therefore, it is reasonable to infer that the sample size was sufficient to effectively capture the gradient effects of these networks.

### 3.2. Gene Expression Associations

The first two components of the PLS regression explained 51.7% of the variance in SCZ-related changes in the primary gradient (component 1: *p* < 0.001; component 2: *p* < 0.05; permutation test with spatial autocorrelation correction, see Appendix A). We computed confidence intervals for the PLS and the results (see Appendix A). Since PLS component 1 and PLS component 2 typically captured the major part of the covariance between X and Y, we chose the first two principal components of the partial least squares for our analysis. The regional mapping of these components showed a positive correlation with the z-map of gradient differences (component 1: r = 0.586, *p* < 0.001; component 2: r = 0.386, *p* < 0.001; Figure 4a,b). The first two principal components of the partial least squares analysis were positively correlated with the difference between individuals with schizophrenia and controls, indicating a significant relationship between the principal components and the difference. We calculated the variable influence on projection (VIP) values of the genes in question, set the threshold to 1, and visualized the genes with VIP values greater than 1 (see Appendix A). The gene enrichment analysis of the functional principal gradient differences (see Appendix A) revealed significant enrichment in bioprocesses related to the modulation of chemical synaptic transmission, synapse organization, and nervous system development. For cellular components, significant enrichment was found in the synaptic parts and dendritic spines (Figure 4c). The genes ordered by the weights of the second principal component of the PLS did not show significant enrichment.

## 4. Discussion

This study investigated the differences in the functional and morphological connectivity gradients between individuals with schizophrenia and healthy controls using a multimodal approach. By integrating neuroimaging and gene expression data, we aimed to provide a comprehensive understanding of the neural alterations underlying schizophrenia.

Our findings indicated that the primary-to-transmodal functional gradient explained a similar proportion of variance in both groups, suggesting that the broad hierarchical organization of brain connectivity is preserved in individuals with schizophrenia. This aligns with previous research highlighting the general stability of large-scale brain organization, even in the context of neuropsychiatric conditions [10,12,40]. However, specific network-level differences were evident. Reduced gradient scores within the visual and sensorimotor networks, alongside increased scores in the limbic network, imply disrupted connectivity that could contribute to the sensory processing deficits and perceptual abnormalities observed in schizophrenia [13,41,42].

The shift in gradient patterns, particularly the decreased differentiation between primary sensory regions and higher order association areas, suggests a breakdown in functional specialization. This reduced separation may impair the integration of sensory inputs with cognitive processes, potentially underpinning symptoms such as hallucinations and cognitive dysfunction [22,43]. The increased scores within the limbic network support the existing evidence of hyperactivity in these regions, which is associated with emotional dysregulation and impaired reward processing [40,41]. Moreover, lower gradient scores in the visual network may underlie difficulties in visual discrimination and attention, which are common in schizophrenia [44]. Similarly, reduced gradient scores in the sensorimotor network may be associated with motor coordination impairments and altered somatosensory perception [45]. These findings indicate a disruption in the functional hierarchy that may contribute to a cascade of effects on higher cognitive functioning. This altered connectivity profile could be linked to difficulties in integrating information across different brain networks, further elucidating the complex clinical presentations of schizophrenia [11,13]. From a clinical perspective, the disrupted connectivity gradients observed in this study hold significant promise as potential biomarkers for schizophrenia. Identifying such biomarkers could enable earlier diagnosis and personalized treatment strategies, as they reflect fundamental neural alterations associated with the disorder. Gradient-based biomarkers may also facilitate the stratification of patients for tailored interventions, such as cognitive remediation or neuromodulation therapies, targeting specific network dysfunctions [46,47]. Additionally, based on the word cloud analysis, the gradients collectively demonstrate that the normal control group exhibited clear functional segregation and balanced network integration, while the schizophrenia group showed a disrupted hierarchy with a shift toward higher order motor planning and sensory dominance, reflecting impaired connectivity and compensatory reorganization, as observed in previous research [20,48].

Although the overall morphological similarity gradients showed no significant group differences, regional variations mirrored the functional findings. Specifically, reduced morphological similarity scores in the visual network and increased scores in the limbic network in schizophrenia suggest that structural alterations parallel functional disruptions [20,27]. This congruence implies a shared basis for functional and structural connectivity changes, emphasizing the notion that structural alterations contribute to the aberrant functional connectivity seen in patients [49,50]. Decreased morphological similarity in the primary sensory areas may reflect developmental or degenerative processes that weaken network specialization, while increased similarity in the limbic regions may indicate compensatory or pathologically heightened connectivity linked to emotional processing and regulation [51,52]. These findings could guide treatment strategies by informing targeted interventions aimed at enhancing functional and structural integration. For instance, cognitive remediation therapies or neurostimulation techniques, such as repetitive transcranial magnetic stimulation (rTMS), could address hyperactivity in limbic regions, potentially alleviating emotional dysregulation [53,54]. Furthermore, structural connectivity findings underscore the potential of neurodevelopmental interventions to strengthen network specialization during critical periods of brain development [55].

The analysis of gene expression data revealed significant associations between specific gene profiles and observed differences in the connectivity gradients. The enrichment of these genes suggests that genetic factors may also influence the development and organization of neural circuits, potentially leading to aberrant connectivity patterns in schizophrenia. The involvement of genes related to neuronal development and synapse formation is particularly noteworthy, as it aligns with the neurodevelopmental hypothesis of schizophrenia [56,57]. Meanwhile, we found that the dendritic spines play a crucial role in synaptic function and plasticity, and their dysregulation has been associated with a range of neuropsychiatric disorders. This suggests that synaptic dysfunction could be a primary factor contributing to the connectivity changes observed in schizophrenia [58,59]. Moreover, the correlation between gene expression and connectivity gradient differences suggests that these genetic factors may directly influence the hierarchical organization of neural circuits. Genes related to axonal targeting and myelination, for example, could affect the long-range connectivity between higher order association regions, leading to the observed compression in connectivity gradients in schizophrenia [60]. This synaptic focus is particularly relevant given that many functional disruptions identified in this study were concentrated in the sensory and limbic networks, areas that depend heavily on synaptic plasticity for efficient information processing [61,62]. The enriched gene expression associated with synaptic and neuronal development suggests a mechanistic link between genetic predispositions and the observed impairments in network integration and specialization in schizophrenia [26,30].

Synaptic dysfunction, driven by both genetic and environmental factors, emerged as a key contributor to connectivity changes in schizophrenia. Mutations in synaptic proteins, such as PSD-95, Shank3, and neurexins, have been linked to synaptic instability and altered neurotransmission. These molecular abnormalities may disrupt the balance between excitatory and inhibitory inputs, impairing the integration of sensory and associative information across brain networks [63,64]. Additionally, deficits in glutamatergic signaling were closely tied to synaptic dysfunction in schizophrenia. This aligns with gene expression findings pointing to the involvement of glutamate-related pathways in connectivity gradient differences [60,65]. Moreover, the compression of the primary-to-transmodal functional gradient and alterations in the visual and limbic networks suggest disruptions in hierarchical brain organization and network integration. Gene enrichment analysis further supported these interpretations, showing the significant involvement of synapse-related pathways, including chemical synaptic transmission and dendritic spine dynamics, which are critical for synaptic plasticity and network stability. These findings align with the neurodevelopmental hypothesis of schizophrenia, where deficits in the synaptic integrity impair the long-range connectivity and cortical organization [63]. Behavioral alterations, such as the cognitive and emotional dysfunctions observed in schizophrenia, may thus emerge from a combination of a reduced sensory processing efficiency and heightened limbic hyperconnectivity, driven by both genetic predispositions and synaptic dysfunction [66]. The impact of potential confounding variables, such as medication effects and comorbidities, should be considered in interpreting these findings. For instance, antipsychotic medications have been shown to influence connectivity patterns, which could partially account for the observed alterations [67,68]. Similarly, comorbid conditions, such as substance use or depression, may further complicate connectivity profiles. To address these concerns, future studies should incorporate larger samples and adjust for these factors to refine our understanding of their role in gradient alterations. Longitudinal studies examining medication-naïve patients could also provide clearer insights into disease-specific connectivity changes [69,70]. These findings offer an initial insight into the connection between these microscopic biological processes and the broader gradient changes observed in schizophrenia.

## 5. Limitations and Future Research

While this study provides valuable insights into the alterations of the functional and morphological connectivity gradients in schizophrenia, it is not without limitations. A primary limitation is the cross-sectional nature of the data, as the database did not provide comprehensive clinical assessments such as cognitive scores. Alternatively, we examined the associations between the gradient alteration maps and meta-analytic cognitive terms from the Neurosynth database [35]. These results should be considered as evidence of an indirect brain–cognition association in schizophrenia. To fully elucidate the temporal dynamics of connectivity alterations, future research should adopt a longitudinal approach, tracking changes in the functional and structural gradients across different stages of the disorder. This would help determine whether the observed alterations precede the symptom onset, progress with the disease severity, or are influenced by treatment effects. Additionally, integrating cognitive assessments in longitudinal studies would clarify how connectivity disruptions relate to cognitive decline or improvement over time.

Another limitation lies in the relatively modest sample size, which may impact the generalizability of the findings. A larger and more diverse cohort would enhance the statistical power and enable more robust subgroup analyses, such as stratifying participants by their symptom severity or treatment status. Additionally, the potential influence of confounding factors, such as medication effects, comorbid conditions, and behavioral factors, must be considered. Although efforts were made to control for these variables, their residual effects cannot be entirely ruled out.

Future studies should address these limitations by incorporating longitudinal designs to examine connectivity alterations over time and their relationship to symptom trajectories. Longitudinal studies, which involve repeated observations of the same individuals over extended periods, allow for the investigation of dynamic changes in brain connectivity and their association with disease progression, the treatment response, and cognitive outcomes. Expanding the sample sizes and ensuring greater demographic and clinical diversity would also strengthen future investigations. Moreover, integrating more targeted gene expression data and cross-modality analyses could provide deeper insights into the biological underpinnings of these connectivity changes. This approach could facilitate the development of biomarkers for early detection and inform the creation of targeted interventions aimed at restoring the functional network integrity in schizophrenia.

## 6. Conclusions

To conclude, our findings provide integrative insights into the functional and structural alterations underlying schizophrenia, highlighting the significant role of genetic factors in shaping these changes. The convergence between functional connectivity and morphological similarity gradients, particularly in the sensory and limbic regions, underscores the intricate relationship between structure and function in the brain of individuals with schizophrenia. The integration of multiparametric neuroimaging and gene expression data offers a more comprehensive view of the disorder, suggesting that both synaptic dysfunction and altered cortical organization contribute to the clinical symptoms of schizophrenia. Future research should focus on the relationships between the genetic risk, brain connectivity alterations, and symptom progression in schizophrenia.

## Figures and Tables

**Figure 1 brainsci-15-00179-f001:**
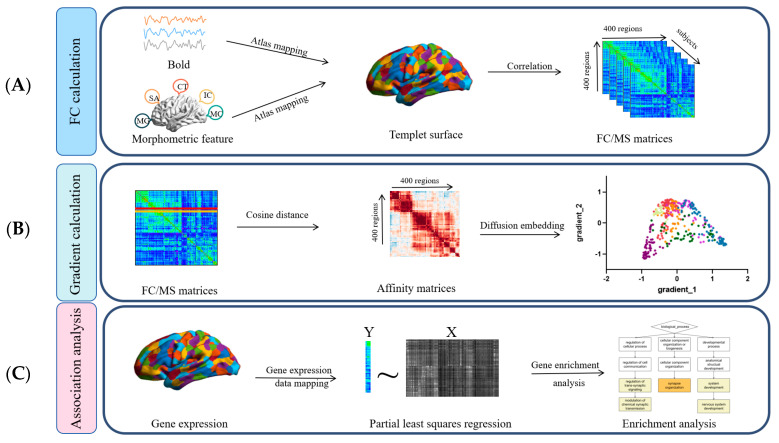
Method framework. (**A**) After functional preprocessing, the functional time series data were extracted from the functional MRI scans, and their Pearson correlations were calculated to obtain the functional connectivity matrix; the morphological features (GM, SA, CT, IC, MC) were calculated from the structural imaging obtained after structural preprocessing, and the morphological similarity matrix was obtained. (**B**) The affinity matrix is obtained by computing the cosine similarity of the connection matrix, and the first two eigenvectors of the affinity matrix are retained by applying diffusion mapping embedding, which corresponds to the primary and secondary gradients, respectively.(**C**) The gene expression data obtained from the AHBA database were set as the independent variable, and the difference in the gradient between controls and individuals with schizophrenia was set as the dependent variable, and the partial least squares method was used to analyze the two, and enrichment analysis was performed on the differentially expressed genes.

**Figure 2 brainsci-15-00179-f002:**
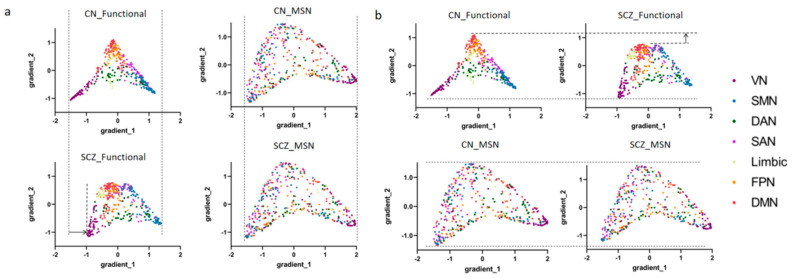
Comparison of functional and morphological similarity gradients between healthy control and schizophrenia groups: (**a**) The primary gradient (gradient_1) comparison: functional connectome hierarchical organization was compressed in individuals with schizophrenia; there was no difference in the morphological similarity gradients between the two groups. (**b**) The secondary gradient (gradient_2) comparison: functional connectome hierarchical organization was compressed in individuals with schizophrenia; there was no difference in the morphological similarity gradients between the two groups.

**Figure 3 brainsci-15-00179-f003:**
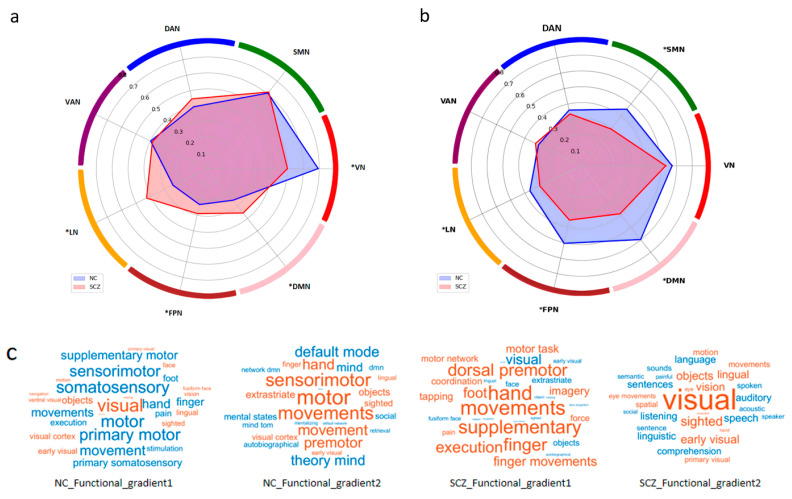
Comparison of functional similarity gradients. * Indicates significant difference between normal control and schizophrenia (*p* < 0.05). (**a**) Radar plots showing the Yeo network profile of each group-level mean functional gradient. Compared with controls, the individuals with schizophrenia scored higher on principal functional gradients in the limbic network, the default mode network, and the prefrontal control network. The visual network had lower functional gradient scores. (**b**) Compared with controls, the individuals with schizophrenia scored lower on secondary functional gradients in the sensorimotor network, the limbic network, the frontoparietal control network, and the default mode network. (**c**) Word clouds representing the top 15 positive (red) and negatively correlated (blue) Neurosynth decoding topic terms for each gradient map of the healthy control and schizophrenia groups; all correlations are significant (FDR q < 0.05).

**Figure 4 brainsci-15-00179-f004:**
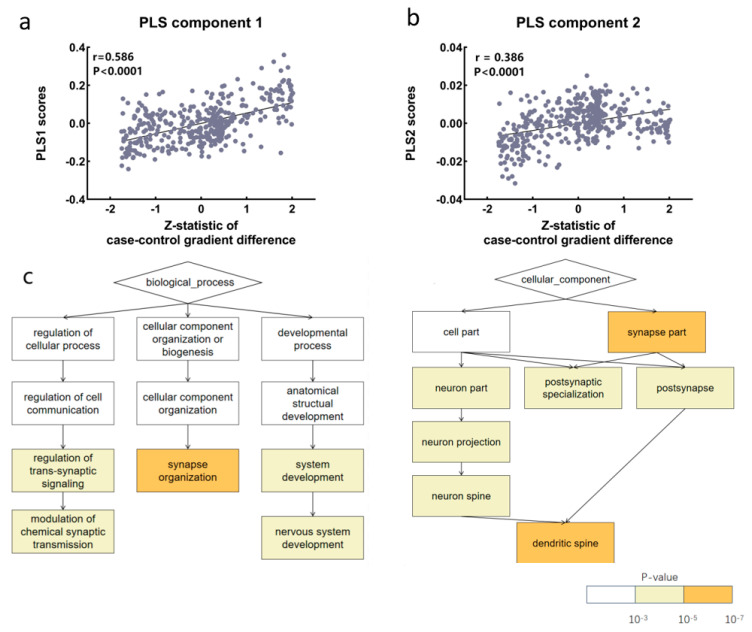
Correlations and gene enrichment analysis related to the functional gradient differences between individuals with schizophrenia and controls. (**a**) Positive correlation between the first principal component of the partial least squares and the difference between individuals with schizophrenia and controls. (**b**) Positive correlation between the secondary principal component of the partial least squares and the difference between individuals with schizophrenia and controls. (**c**) Principal functional gradients of z-map are significantly enriched in biological processes in the regulation of chemical synaptic transmission, synaptic organization, and nervous system development. Significant enrichment in the synapse parts and dendritic spines for cellular components.

**Table 1 brainsci-15-00179-t001:** Summary of COBRE dataset subjects.

Group	Control	Schizophrenia
Number of people	74	72
Age (mean ± standard deviation)	35.87 ± 11.74	38.16 ± 13.89
Sex (male/female)	51/23	58/14

## Data Availability

The original contributions presented in this study are included in the article and Appendix A. Further inquiries can be directed to the corresponding author.

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
