# Peer review of "Investigating Connectivity Gradients in Schizophrenia: Integrating Functional, Structural, and Genetic Perspectives"

_brainsci, 2025, doi:10.3390/brainsci15020179_

Round 1

Reviewer 1 Report

Comments and Suggestions for Authors

This research article investigates brain connectivity disruptions in schizophrenia using a multimodal approach. Functional and structural MRI data, along with gene expression data, were analyzed from individuals with schizophrenia and healthy controls. The study found altered connectivity gradients in schizophrenia, particularly in visual/sensorimotor and limbic networks, linked to genes involved in synaptic signaling and neuronal development. These findings suggest widespread network-level abnormalities in schizophrenia with a strong genetic component. The researchers propose that their findings could contribute to the development of biomarkers and treatments.

The paper is well-structured, and I have some suggestions to further improve the manuscript.

1. Kindly change "2.2. Image Acquisition and Preprocessing"  to '2.2. Image Processing'. The "2.1. Participants" should focus only on subject data. Kindly allocate and write a separate part for "Data Acquisition"

2. I recommend authors use 'multiparametric' instead of "multimodal" in some areas that exact multiple quantitative parameters from specific MRI data. Otherwise, you can use the "multimodal approach"

3. The justification for retaining the top 10% of connections for each ROI in the functional connectivity analysis requires further explanation. Kindly cite the "Generating gradients" part of the BrainSpace study.

4. The selection of the first two principal components of the PLS regression for further analysis needs more justification. "Obtain the first two principal components of the partial least squares."

5. While the authors mention comparing gradient scores between groups using paired t-tests, specifying the precise tests employed for different brain networks would improve clarity. 

6. Table 1 demographic data can be further supplemented with additional information related to functional and cognitive scores.

Minor  point: If you define a phrase such as "...partial  least squares (PLS)." Kindly use PLS in another use (line 157).

Reviewer 2 Report

Comments and Suggestions for Authors

This research paper investigates connectivity gradients in schizophrenia, integrating functional, structural, and genetic perspectives. While the study presents interesting findings, several improvements are needed to enhance its rigor, clarity, and overall persuasiveness.

Strengths:

Multimodal Approach: the integration of fMRI, sMRI, and gene expression data is a significant strength. This multimodal approach offers a more comprehensive understanding of the complex interplay between brain structure, function, and genetics in schizophrenia.

Novel Application of Connectivity Gradients: the use of connectivity gradients to analyze brain networks provides a novel perspective, potentially revealing subtle yet important alterations in network organization that might be missed by traditional methods.

Gene Expression Analysis: the incorporation of gene expression data adds a crucial layer of biological insight, linking connectivity changes to potential molecular mechanisms.

Suggested improvements:

1.  Sample Size and Power: the sample size (146 participants) is relatively modest for a study of this complexity.  A larger sample, especially with a more balanced distribution across schizophrenia subtypes, is crucial to increase statistical power and to reduce the risk of type II errors.  The power analysis should be explicitly stated and justified. Conduct a thorough power analysis to justify the sample size. 

2.  Clinical Correlates: the study lacks detailed clinical data.  Integrating comprehensive clinical assessments (e.g., symptom severity scales like PANSS, cognitive test scores, etc.) would allow for crucial correlation analyses to explore the relationships between specific connectivity patterns and clinical manifestations.  This is essential for establishing clinical relevance and improving the paper's impact.

3.  Statistical Methodology: while PLS regression is used, a more comprehensive statistical strategy is needed.  This includes detailed descriptions of all statistical methods used, justifications for their selection, and robust handling of multiple comparisons.  Consider reporting effect sizes and confidence intervals in addition to p-values.  Moreover, exploring alternative statistical approaches (e.g., more advanced machine learning methods) could provide additional insights.

4.  Control Group Selection:  the description of the control group should be more detailed.  Were controls carefully matched to the patient group on relevant demographic and clinical factors (age, sex, education level)?  This information is vital to ensure comparability.

5.  Interpretation of Results: the interpretation of specific findings needs to be more nuanced.  For instance, while the study notes compression in the main functional gradient, the biological mechanisms underlying this compression are not thoroughly discussed.  More detailed mechanistic explanations linking genetic variations, connectivity changes, and specific clinical symptoms are needed.

6.  Discussion of Limitations: the discussion of limitations is relatively brief.  A more comprehensive and critical assessment of the study's limitations (sample size, cross-sectional design, potential confounders) would strengthen the paper's credibility.

7.  Clarity and Structure: certain sections of the paper could benefit from improved clarity and organization.  The flow of the manuscript could also be improved to enhance readability and persuasiveness. 

Comments on the Quality of English Language

Use concise and precise language. Carefully revise the manuscript for grammar and style

Round 2

Reviewer 1 Report

Comments and Suggestions for Authors

Thank you for addressing my comments. I have no additional comment.

Author Response

Thank you very much for your evaluation of our manuscript. With your help, we have further improved the content of our manuscript and made its structure clearer.

Reviewer 2 Report

Comments and Suggestions for Authors

The introduction provides a thorough overview of schizophrenia, emphasizing its complexity and the gaps in understanding the relationship between brain structure and function. The authors effectively highlight the importance of connectivity gradients as a framework for exploring neural disruptions in schizophrenia. The literature review is comprehensive, drawing on relevant studies to justify the need for a multiparametric approach. However, the introduction could be improved by briefly discussing the potential clinical implications of the study, such as its relevance for treatment or diagnosis. The study design is robust and aligns with the stated objectives. The integration of structural MRI (sMRI), functional MRI (fMRI), and gene expression data is appropriate for examining connectivity gradients in schizophrenia. The use of the COBRE dataset ensures a sufficient sample size for statistical validity. The inclusion of healthy controls strengthens the comparative aspect of the study. However, the cross-sectional nature of the design limits the ability to infer causal relationships. Future research could benefit from a longitudinal approach to track changes over time.  The methods are well-documented and comprehensive. The authors clearly describe the participant selection process, imaging parameters, and preprocessing steps. The use of advanced tools such as the DPABI toolbox, BrainSpace, and abagen for gene expression analysis demonstrates methodological rigor. The statistical analyses, including partial least squares regression and enrichment analysis, are appropriate for the research questions. However, additional details on how confounding factors, such as medication and comorbid conditions, were managed would enhance the reproducibility and reliability of the study. The results are presented clearly and supported by detailed visualizations, including gradient maps and radar plots. The authors effectively demonstrate the differences in functional and morphological connectivity gradients between individuals with schizophrenia and healthy controls. The findings of reduced functional differentiation in sensory and associative regions and increased connectivity in limbic areas provide valuable insights into the neural basis of schizophrenia. The gene expression analysis adds depth, linking genetic factors to observed connectivity patterns. While the results are compelling, a more detailed exploration of the clinical significance of these findings would strengthen their impact. The discussion thoughtfully interprets the results, integrating them with existing literature on schizophrenia. The authors provide a convincing argument for the role of disrupted connectivity gradients and genetic influences in the disorder. The emphasis on synaptic dysfunction and its implications for brain network organization is particularly noteworthy. The conclusion succinctly summarizes the study’s contributions and highlights future directions, such as the need for longitudinal studies and larger, more diverse cohorts. However, the discussion could benefit from a more explicit consideration of how these findings could inform clinical practice, such as identifying biomarkers for early diagnosis or targets for therapeutic interventions.

Tips:

  1. Discuss potential clinical implications, such as the use of connectivity gradients as biomarkers.

  2. Address the impact of confounding variables, such as medication and comorbidities, in greater detail.

  3. Consider expanding the discussion on how the findings relate to symptomatology and treatment strategies.

  4. Incorporate a plan for longitudinal research to explore connectivity changes over time.

Author Response

Thank you very much for your feedback on our manuscript. Based on your suggestions, we have added the clinical significance and molecular related content of the study in the introduction section, And added potential confounding variables and clinically relevant content in the discussion section. This modification makes our manuscript more complete and easier to understand.